# Critical Quality Control Methods for a Novel Anticoagulant Candidate LFG-Na by HPSEC-MALLS-RID and Bioactivity Assays

**DOI:** 10.3390/molecules27144522

**Published:** 2022-07-15

**Authors:** Shunliang Zheng, Yi Wang, Jiashuo Wu, Siyao Wang, Huaifu Wei, Yongchun Zhang, Jianbo Zhou, Yue Shi

**Affiliations:** 1Institute of Medicinal Plant Development, Chinese Academy of Medical Sciences and Peking Union Medical College, Beijing 100193, China; zhengshunliang@163.com (S.Z.); wjs_implad@163.com (J.W.); 2Mudanjiang Youbo Pharmceutical Co., Ltd., Mudanjiang 157013, China; realwangyi1221@163.com (Y.W.); wsiy11@126.com (S.W.); whf8369@163.com (H.W.); ybyyzyc@163.com (Y.Z.)

**Keywords:** LFG-Na, sea cucumber, *Holothuria fuscopunctata*, molecular weight, HPSEC-MALLS-RID, bioactivity assay, quality control

## Abstract

A low molecular weight fucosylated glycosaminoglycan sodium (LFG-Na) is a novel anticoagulant candidate from the sea cucumber *Holothuria fuscopunctata* that selectively inhibits intrinsic tenase (iXase). The molecular weight, molecular weight distribution and bioactivities are the critical quality attributes of LFG-Na. The determination of these quality attributes of such an oligosaccharides mixture drug is challenging but critical for the quality control process to ensure its safety and efficacy in clinical use. Herein, the molecular weight and molecular weight distribution of LFG-Na were successfully determined using high performance size exclusion chromatography coupled with multi angle laser light scattering and refractive index detector (HPSEC-MALLS-RID). Comparing to the conventional method, HPSEC-MALLS-RID based on the refractive index increment (*dn/dc*) did not require the reference substances to establish the calibration curve. The acceptance criteria of LFG-Na were established, the weight-average molecular weight ($M_{w}$) should be 4000 to 6000 Da, the polydispersity ($M_{w}$/$M_{n}$) < 1.40, and the fraction with molecular weights of 1500 to 8000 Da should be no less than 80% of the total. HPSEC-MALLS-RID was also utilized for the determination of the starting material native fucosylated glycosaminoglycan (NFG) to choose a better manufacturing process. Furthermore, APTT assay was selected and the potency of anti-iXase, referring to the parallel line assay (PLA) method, was established to clarify the consistency of its biological activities. The results suggest that HPSEC-MALLS-RID and bioactivity assays are critical quality control methods for multi-component glycosaminoglycan LFG-Na. The methods also provide a feasible strategy to control the quality of other polysaccharide medicines.

## 1. Introduction

Venous thromboembolism (VTE), such as deep vein thrombosis (DVT) and pulmonary embolism (PE), is a common underlying pathology of cardiovascular disease, which is a global health burden associated with high morbidity and mortality [1,2]. Anticoagulants, antiplatelets and thrombolytics are three types of antithrombotic drugs in great demand [3]. Among them, anticoagulants are effective in inhibiting the activity or synthesis of coagulation factors, which ultimately prevent or limit the formation of fibrin clots by breaking the coagulation cascade.

To date, numerous anticoagulant drugs have been developed. The classical management of VTE in adults consists of an initial treatment with adjusted-dose intravenous unfractionated heparin (UFH), body weight-adjusted subcutaneous low molecular weight heparin (LMWH), or body weight-adjusted subcutaneous fondaparinux followed by long-term treatment with a vitamin K antagonist (VKA) [4]. UFH and LMWH have been the clinical cornerstones of antithrombotic treatment and prophylaxis for over 80 years, and early in the 21st century, direct oral anticoagulants (DOACs) targeting thrombin (f.IIa) or factor Xa (f.Xa) were developed. However, the risk of haemorrhagic complications is still a major concern with their clinical application, and their therapeutic monitoring requirement is controversial [5,6,7,8]. Consequently, there is an unmet medical need in discovering safer anticoagulants for antithrombotic therapy. 

The inhibitors of intrinsic coagulation pathway can inhibit pathological thrombosis without or slightly affecting hemostatic function and prevent thrombosis with negligible bleeding risks [9,10]. Intrinsic factor Xase complex (iXase) consisting of f.IXa-f.VIIIa is the last and rate-limiting enzyme of the intrinsic coagulation pathway [10,11,12], indicating that targeting iXase is a promising safer anticoagulant therapy with lower risk of bleeding. 

Native fucosylated glycosaminoglycan (NFG) from sea cucumber is a unique glycosaminoglycan with fucose branches. The NFG extracted from the sea cucumber *Holothuria fuscopunctata* Jaeger mainly exhibits a chondroitin sulfate (CS) chain and 3,4-di-*O*-sulfated-fucose (Fuc_3S4S_) branches [10,13]. Its β-eliminative depolymerized product, a low molecular weight fucosylated glycosaminoglycan sodium (LFG-Na), is a novel anticoagulant candidate to enter clinical trials permitted by the U.S. Food and Drug Administration. LFG-Na has clear chemical composition, selective anti-iXase activity, potent anticoagulation, antithrombosis with low bleeding tendency and predictable pharmacodynamic characteristics without the NFG’s undesired effects of platelet aggregation and factor XII (f.XII) activation [12,14].

The biological and pharmacological effects of fucosylated glycosaminoglycans are closely related to their molecular weight, molecular weight distribution and sulfation patterns [15,16]. LFG-Na is composed of a series of oligosaccharides, and it is thus a complex multicomponent drug. Moreover, its pharmacological activities, such as anti-iXase, factor IXa-binding, anticoagulant and antithrombotic activities, result from its oligosaccharides in the terms of weighted average sum [12]. Therefore, the molecular weight, molecular weight distribution and bioactivities are the critical quality attributes of LFG-Na. 

The current typical method for the determination of the molecular weight and molecular weight distribution of glycosaminoglycans is high performance gel permeation chromatography (HPGPC) [12,17]. The data are calculated using GPC software, so it is necessary to fit the calibration curve with a series of narrow standard reference standards with known molecular weight for calculation. Currently, it is temporarily impossible to obtain a series of narrow standard references that completely cover the maximum molecular weight range of glycosaminoglycans, including NFG and LFG-Na, due to the complicated and laborious procedures for their separation and purification [18]. The accuracy of HPGPC with standard curves is relatively poor. Additionally, it is time consuming, considering the injection of the standard references. Therefore, accurate and rapid determination of the molecular weight of LFG-Na and its starting material NFG is crucially important and urgently required to control quality of the new drug. Herein, high performance size exclusion chromatography coupled with multi angle laser light scattering and refractive index detector (HPSEC-MALLS-RID) was used to determine the molecular weight and molecular weight distribution of LFG-Na and compared to HPGPC. Furthermore, the chemical characteristics and bioactivities of multiple batches of LFG-Na were tested and compared to their consistency. Our results indicate that HPSEC-MALLS-RID and bioactivity assays are critical quality control methods for the multi-component anticoagulant candidate LFG-Na.

## 2. Results and Discussion

### 2.1. Determination of the Refractive Index Increment (dn/dc)

For the multi-angle light scattering coupled with size exclusion chromatography (SEC-MALLS), the *dn/dc* of the solution is required. The *dn/dc* of a solution is a constant that indicates the variation of the refractive index with the solute concentration. It is used in the multi-angle light scattering technique to determine the concentration and the weight-average molecular weight ($M_{w}$) of polymers [19]. Since the *dn/dc* appeared as an important parameter, the accurate value was therefore essential for the determination of the $M_{w}$ of LFG-Na and its starting material NFG.

A representative chromatogram obtained from the determination of the *dn/dc* value of LFG-Na is shown in Figure 1, whereas the chromatograms of NFG samples are similar. The changes in concentration of glycosaminoglycan solutions were converted to changes in refractive index. The *dn/dc* values were obtained with ASTRA software (version 7.1.3, Wyatt Technology Co., Santa Barbara, CA, USA).

Table 1 shows *dn/dc* values of LFG-Na and NFG in the solution of 0.1 mol/L sodium nitrate (containing 0.02% sodium azide). Five batches of LFG-Na samples had *dn/dc* values varying from 0.1173 to 0.1283 mL/g, with a mean value of 0.1248 mL/g. Six batches of NFG samples had *dn/dc* values varying from 0.1113 to 0.1192 mL/g, with a mean value of 0.1163 mL/g; while four batches of NFG-2.0M (obtained before the NFG manufacturing process optimization) samples had a mean *dn/dc* value of 0.1166 mL/g. The mean *dn/dc* value of LFG-Na was slightly larger than that of NFG. These mean *dn/dc* values of LFG-Na and NFG were used for the calculation of their $M_{w}$ by the method of HPSEC-MALLS-RID.

### 2.2. The Starting Material NFG, NFG-2.0M and Their Molecular Weights

Six batches of NFG were extracted and purified from dried body wall of the sea cucumber *H. fuscopunctata* Jaeger. The yields of NFGs were about 0.85% by dry weight and the yields of another four batches of NFG-2.0M were about 1.20% by dry weight.

After being subjected to a Shodex Ohpak SB-804 HQ column, the HPSEC-MALLS-RID of NFG showed only one peak, while NFG-2.0M showed two peaks (Figure 2). As shown in Figure 2B, NFG-2.0M appeared to contain NFG (peak 2) and a small amount of sulfated fucan SF-II (peak 1) according to the studies reported previously [20]. The $M_{w}$ and the polydispersity ($M_{w}$/$M_{n}$) of NFG and NFG-2.0M by HPSEC-MALLS-RID (only the NFG peak was involved in the calculation) are presented in Table 2. The $M_{w}$ of NFG from different batches were determined to range from 58,270 to 74,280 Da, while that of NFG-2.0M were larger (from 77,680 to 86,950 Da). The $M_{w}$/$M_{n}$ of NFG from different batches were determined to range from 1.105 to 1.241, while that of NFG-2.0M were higher (from 1.297 to 1.407). Furthermore, the labeled value (true value) of $M_{w}$ of BSA was 66,430 Da, and the mean value of $M_{w}$ from the determinations was 64,100 (the RSD was 0.67%, n = 5). The standard deviation is 3.51% of the true value of $M_{w}$, which indicates that the HPSEC-MALLS-RID has good accuracy and system suitability. The RSD of repeatability for NFG is 1.6% (n = 6), which suggests that the method for NFG has good repeatability. Meanwhile, the RSD of stability (0 h, 4 h, 8 h, 12 h, 24 h at room temperature) for NFG solutions was 1.27% (n = 5), which indicates that NFG was stable in 0.1 mol/L sodium nitrate (containing 0.02% sodium azide) at room temperature during the tested period. It was found that the $M_{w}$ and the $M_{w}$/$M_{n}$ of the NFG peak (peak 2) of the NFG-2.0M sample could be not precisely determined due to the relatively poor resolution of the column and the co-elution of the sulfated fucan SF-II (peak 1) [20].

These data indicate that the purity and homogeneity of NFG were better than NFG-2.0M after the manufacturing process optimization. That is to say, the FPA98 ion-exchange resin elution process was the crucial step for ensuring the purity and homogeneity of NFG. As the starting material, NFG was more conducive to the quality control of LFG-Na. Therefore, the manufacturing process after optimization of NFG was selected. Based on multiple batches of measurement data, the acceptance criteria of the molecular weight of NFG were established. Considering that NFG was just the starting material of LFG-Na, the acceptance criteria could be relatively broad. Therefore, the $M_{w}$ of NFG should be 50,000 to 10,000 Da, and $M_{w}$/$M_{n}$ < 1.50.

### 2.3. Determination of Molecular Weight and Molecular Weight Distribution of LFG-Na Using HPSEC-MALLS-RID

HPSEC-MALLS-RID, an absolute method, has been proven as the powerful and efficient technique for analysis of the molecular weight and molecular weight distribution of polysaccharides in dilute polymer solution without using a series of standards [21]. In some countries, molecular weight has been considered as one of the indicators to control the quality of some drugs in pharmacopoeias, such as LMWH [22]. Therefore, the established HPSEC-MALLS-RID method was applied for the determination of the molecular weight and molecular weight distribution of LFG-Na.

Five batches of LFG-Na were prepared from NFG by β-eliminative depolymerization. The representative HPSEC-MALLS-RID chromatogram of LFG-Na is shown in Figure 3. Only one fraction of glycosaminoglycan was detected (dRI), whereas the peak of retention time from 22 to 24 min in the HPSEC-MALLS-RID chromatogram was the solvent peak. The result showed that LFG-Na was distributed from 15 to 19 min. Since there was no peak between the retention time of 10 to 15 min in the RI chromatogram (dRI), even if a peak was found in the MALLS chromatogram (LS), it could be considered that there was no substance from 10 to 15 min based on the different detection principles of MALLS and RI detectors.

Using the Astra software (version 7.1.3, Wyatt Technology Co., Santa Barbara, CA, USA)), the results of molecular weight and molecular weight distribution for LFG-Na by HPSEC-MALLS-RID are summarized in Table 3. The $M_{w}$ of LFG-Na from different batches were determined to range from 4596 to 5708 Da; it could be considered that the $M_{w}$ of LFG-Na from different batches were similar. Sample L1 was observed as the lowest $M_{w}$ among the test samples, while sample L5 was the highest one. The true value of $M_{w}$ of BSA was 66,430 Da, and the mean value of $M_{w}$ from the determinations was 64,100 (the RSD was 0.67%, n = 5). The standard deviation was 3.51% of the true value of $M_{w}$, which indicated that the HPSEC-MALLS-RID had good accuracy and system suitability. The RSDs of repeatability and intermediate precision for the $M_{w}$ of LFG-Na were 2.13% (n = 6), and 2.87% (different days, n = 12), respectively. The data suggest that the HPSEC-MALLS-RID for LFG-Na has good accuracy, system suitability, repeatability and intermediate precision. The molecular weight distribution is used to measure the width of the molecular weight distribution, which represents the homogeneity and dispersibility of the polysaccharides. The $M_{w}$/$M_{n}$ of LFG-Na from different batches were determined to range from 1.121 to 1.240, which suggest that the molecular weight distribution of each batch of LFG-Na was relatively narrow. Furthermore, the percentages of the fractions of LFG-Na with different molecular weights were obtained (Table 3). From the data of five batches of LFG-Na, the percentages of the fraction with molecular weights range from 1500 to 8000 Da (M1500~8000) were all more than 85%, and the percentages of the fraction with molecular weights greater than 8000 Da (M8000) were all less than 15%, while there was no fraction with molecular weights lower than 1500 Da (M1500).

Based on multiple batches of measurement data, the acceptance criteria of the molecular weight and molecular weight distribution of LFG-Na were established. The $M_{w}$ of LFG-Na should be 4000 to 6000 Da, $M_{w}$/$M_{n}$ < 1.40, the fraction with molecular weights of 1500 to 8000 Da should be no less than 80% of the total, and the fraction with molecular weights greater than 8000 should be no more than 20% of the total.

### 2.4. Determination of Molecular Weight and Molecular Weight Distribution of LFG-Na Using HPGPC

Size exclusion chromatography has been widely employed for separation of polysaccharides and the molecular weight and molecular weight distribution could be determined by HPGPC with suitable standards. Five batches of LFG-Na were also determined by HPGPC with five oligosaccharide standards HS5, HS8, HS11, HS14 and HS17. The HPGPC profiles of the representative batch of LFG-Na and oligosaccharide standards (Figure 4A) show that LFG-Na is the mixture of oligosaccharides with different degrees of polymerization. 

The $M_{w}$ of LFG-Na was calculated according to the calibration curve from the data of oligosaccharide standards and the distribution plot of LFG-Na (Figure 4B) was analyzed using Agilent GPC/SEC software (version A.02.01, Agilent Technologies, Palo Alto, CA, USA). The results of molecular weight and molecular weight distribution for LFG-Na by HPGPC were also summarized in Table 3. The $M_{w}$ of LFG-Na from different batches were determined to range from 5082 to 5657 Da. Sample L1 was observed as having the highest $M_{w}\;$ among the test samples, while sample L4 had the lowest one. The $M_{w}$/$M_{n}$ from different batches were determined to range from 1.280 to 1.438. From the data of five batches of LFG-Na by HPGPC, the percentages of the fraction of M1500, M1500~8000, and M8000 were <3%, >80% and <17%, respectively. The above results suggest that the molecular weight and molecular weight distribution of LFG-Na from different batches were similar.

### 2.5. Comparison of the Determination Using HPGPC and HPSEC-MALLS-RID

The analysis results of molecular weight and molecular weight distribution for LFG-Na using the developed HPSEC-MALLS-RID method were also compared with those of HPGPC analysis. As shown in Table 3 and Figure 5, the $M_{w}$ of LFG-Na from different batches using the HPSEC-MALLS-RID method were relatively close to those of HPGPC analysis (Figure 5A), though the measured value of Sample L1 had a largest absolute error of 1061 Da as compared with that of the HPGPC method (4596 Da vs. 5657 Da), while the measured value of Sample L5 had a smaller absolute error of 177 Da (5708 Da vs. 5531 Da). However, Figure 5B shows that the $M_{w}$/$M_{n}$ of LFG-Na from different batches using HPSEC-MALLS-RID were all obviously smaller than those obtained by HPGPC (average of 1.17 vs. 1.36). There were slight differences in the percentages of the fraction of M1500, M1500~8000 and M8000 between the two methods. The percentages of the fraction of M1500~8000 were higher by the HPSEC-MALLS-RID method. 

In conclusion, both HPSEC-MALLS-RID and HPGPC methods for the determination of LFG-Na included size exclusion chromatography, but were calculated in different ways. HPGPC was widely used and convenient, but the choice of different standards apparently affected the molecular weight result and the standards’ structures should be similar to the samples for obtaining accurate and reliable results. In this study, because the highest molecular weight of oligosaccharide standards was 5328 Da (HS17), which did not cover the maximum molecular weight range of LFG-Na, the accuracy of HPGPC was poor. In addition, in view of the current technical means, it was temporarily impossible to obtain a series of narrow oligosaccharide standards that completely covers the maximum molecular weight range of LFG-Na due to the complicated and laborious procedures for their separation and purification. Compared to HPGPC, the method of HPSEC-MALLS-RID based on the *dn/dc* did not require the reference substances to establish the calibration curve, and the molecular weight and molecular weight distribution of NFG and its depolymerized fraction LFG-Na could be determined directly, efficiently and accurately. The limitation of the HPSEC-MALLS-RID method is that the *dn/dc* value of the sample should be determined or obtained before determining the molecular weight and molecular weight distribution. In addition, the accuracy and system suitability of the method should be assessed by standards, such as BSA or Dextran, which were used at the beginning of each run sequence.

### 2.6. Comparison of the Chemical Characteristics of LFG-Na

The monosaccharide composition analysis with the method of PMP (1-Phenyl-3-methyl-5-pyrazolone) precolumn derivatization-HPLC showed that LFG-Na was composed of glucuronic acid, N-acetyl galactosamine, and fucose (Figure 6). Five batches of LFG-Na (L1-L5) were also analyzed by the Similarity Evaluation System for Chromatographic Fingerprint of TCM software (Version 2012, Chinese Pharmacopoeia Commission, Beijing, China). The HPLC fingerprint similarities were more than 0.99, which showed the perfect correlation and similarity among them. 

NMR spectroscopy was performed using a 600 MHz spectrometer, and the ^1^H/^13^C NMR chemical shifts and NMR spectra of LFG-Na were analyzed. The complete assignment of its signal peaks is given in Table 4. In the ^1^H NMR spectrum, a strong signal at 5.760 ppm could be assigned to the proton H-4 of Δ4,5-unsaturated glucuronic acid (ΔU). There were three relatively strong signal peaks (at 5.356, 5.282, 5.116 ppm) from 5.0 to 5.6 ppm, which were the α-fucose terminal proton signals at different sites with the sulfation type of Fuc_3S4S_. It could be judged in combination with the other spectra that the signals at 5.356, 5.282, and 5.116 ppm are connected to the glucuronic acid, the unsaturated glucuronic acid at the non-reducing end, and Fuc_3S4S_ terminal hydrogen of the GlcA-ol at the reducing end, respectively. The β-terminal proton signals of the main chain appeared at about 4.3 to 4.6 ppm and GlcA-ol was located at the reducing end with the proton H-1 (CH_2_) at 3.800 and 3.761 ppm. Fuc methyl proton signals showed different chemical shifts due to their different locations. Specifically, the Fuc methyl protons at the reducing end (rF) and non-reducing end (dF) occurred at 1.30 to 1.35 ppm, Fuc in the sugar chain, connecting with glucuronic acid (F), occurred at about 1.4 ppm; and GalNAc acetyl protons occurred at about 1.9 to 2.1 ppm. In the ^13^C NMR spectrum, the C-1 (dU and dA) peaks of GlcA and GalNAc at the non-reducing end of the main chain appeared at about 105.91 and 102.51 ppm, the C-1 (U and A) peaks of GlcA and GalNAc in the middle of the main chain appeared at about 106.71, and 102.34 ppm, and the C-1 (rU and rA) peaks of glucuronic acid alcohol and GalNAc at the reducing end appeared at 65.40 and 104.33 ppm, indicating the β-configuration of the bridgehead hydrogen was consistent with the GlcA-ol at the reducing end. C-1 of Fuc connected to different sites also showed different chemical shifts (dF:101.15 ppm, F:102.06 ppm and rF:104.33 ppm). C-3, C-4, and C-5 of ΔU at the non-reducing end appeared at 79.90, 109.54, and 149.79 ppm, respectively. With C-6 (rU) of GlcA-ol at reducing end and C-6 (ΔU) at non-reducing end ΔU appearing at 180.22 and 171.76 ppm as the exception, the carbonyl carbon C-6 (U6) in the GlcA carboxyl and the carbonyl carbon C-7 (rA/A/dA) in the GalNAc acetamido appeared at almost the same site (~177 ppm); Fuc methyl carbon signals appeared at about 18 to 19 ppm; the methyl carbon signal in GalNAc acetamido appeared at about 25 ppm. The C-2 signal of GalNAc appeared at about 54 ppm. The structure of LFG-Na is confirmed as shown in Figure 7A, which is consistent with our previous reports [10,12].

As shown in Figure 6 and Figure 7B,C, HPLC profiles of PMP derivatives and 1D (^1^H/^13^C) NMR spectra with the typical signal assignments from five batches of LFG-Na (L1-L5) were obtained. After comparing and evaluating carefully, the chemical characteristics of these five batches of LFG-Na were highly similar to each other. 

### 2.7. Comparison of Biological Potency

The activity of FG derived from various sea cucumber species has been reported [10,23]. Herein, the APTT (activated partial thromboplastin time) assay was selected and the potency for anti-iXase, referring to the PLA (parallel line assay) method, was established. The APTT-prolonging activity of LFG-Na in human plasma indicated the potent inhibitory activity on the intrinsic coagulation pathway. The inhibition of APTT by LFG-Na was increased depending on the increasing concentration of LFG-Na in a certain range, and the bioactivity could be measured by measuring the concentrations of LFG-Na required to double APTT. The PLA method for estimating the potency for anti-iXase was established and validated with respect to specificity, linearity and range, repeatability, intermediate precision, accuracy and durability in our laboratory.

The activities of multiple batches of samples of LFG-Na are summarized in Table 5. Concentrations required to double the APTT of five batches samples were from 7.36 to 8.89 μg/mL. Sample L4 and L5 with higher $M_{w}$ showed slightly stronger anticoagulant activity than L1-L3. In order to compare the potency for anti-iXase among different batches of LFG-Na, L5 with the highest $M_{w}$ was selected as the reference substance, and the potency for anti-iXase was set as 100 U/mg. The results of potency for anti-iXase also indicated that L4 and L5 had slightly stronger activities of anti-iXase, whereas potency for anti-iXase of L1-L3 were 88.7, 85.9 and 89.8 U/mg. Basically, different batches of LFG-Na with the $M_{w}$ ranging from 4000 to 6000 Da had similar activities, and the results of bioactivity assays in vitro indicated the stability and consistency of the preparation process of LFG-Na. Overall, the bioactivity assays also could be listed as the quality control methods of multi-component drugs to ensure the consistency of their efficacy in clinical use. 

## 3. Materials and Methods

### 3.1. Materials and Chemicals

Dried sea cucumbers *H. fuscopunctata* Jaeger were purchased from local markets in Guangdong Province and Hainan Province, China. Five oligosaccharide standards from LFG-Na (HS5, pentasaccharide, $M_{w}$ = 1506 Da; HS8, octasaccharide, $M_{w}$ = 2462 Da; HS11, hendecasaccharide, $M_{w}$ = 3417 Da; HS14, tetradecasaccharide, $M_{w}$ = 4373 Da; HS17, heptadecasaccharide, $M_{w}$ = 5328 Da) were obtained from Kunming Institute of Botany, Chinese Academy of Sciences. BSA standard was purchased from Sigma (St. Louis, MO, USA). The monosaccharides including L-fucose and *N*-acetyl-D-galactosamine were purchased from Sigma-Aldrich, and D-glucuronic acid was from J&K Scientific Ltd. (Beijing, China). 1-Phenyl-3-methyl-5-pyrazolone (PMP) was purchased from Xiya Reagnent Co. (Linyi, China). Amberlite FPA98Cl ion-exchange resin was purchased from Rohm and Haas Company (Philadelphia, PA, USA). Deuterium oxide (D_2_O, 99.9% Atom D) was obtained from Sigma-Aldrich. The activated partial thromboplastin time (APTT) kits, CaCl_2_ and standard human plasma were purchased from MDC Hemostasis (Neufahrn, Germany), BIOPHEN FVIII: C kit was from Hyphen Biomed (Neuville sur Oise, France). Human factor VIII was from Bayer HealthCare LLC. Deionized water was prepared by the Millipore Milli Q-Plus system (Millipore, Billerica, MA, USA). All other reagents were of analytical grade and obtained commercially.

### 3.2. Preparation of the Starting Material NFG

The NFG was extracted and purified from the sea cucumber *H. fuscopunctata* using a previously described procedure with minor modifications [12,24]. Briefly, the tissue of the dried body wall of the sea cucumber (30 kg) was digested by 1% papain (6000 U/mg) for 6 h and the polysaccharide components were released by 0.25 mol/L sodium hydroxide for 2 h. The crude polysaccharide was obtained by repeated salting-out with KOAc and precipitation by ethanol, which was further purified by strong anion-exchange chromatography using FPA98 ion-exchange resin (30 cm × 160 cm, 100 L resin), and the sample was eluted sequentially with distilled water, 0.5, 1.0 and 1.3 mol/L NaCl solutions. The fraction eluted by 1.3 mol/L NaCl solutions was collected, desalted by a Pellicon device with a 0.5 m^2^ PES membrane cassette with a molecular weight cutoff (MWCO) of 10 kDa (Millipore) and lyophilized to obtain white powder. The NFG was the starting material, which was further used for chemical cleavage to prepare its low molecular weight fragments named ‘LFG-Na’. Before expanding to pilot scale, six batches of NFG were obtained from the dried body wall of the sea cucumber (30 kg) according to the above procedures, based on which the FPA98 ion-exchange resin elution process was optimized.

Before the NFG manufacturing process optimization, we obtained another four batches of NFG (named ‘NFG-2.0M’), which were from the fraction eluted by 2.0 mol/L NaCl solutions during the FPA98 ion-exchange resin elution process. In other words, after going through the processes of enzymolysis, alkaline hydrolysis, salting-out and alcohol precipitation, the crude polysaccharide was eluted sequentially with distilled water, 0.5, 1.0 and 2.0 mol/L NaCl solutions; then the fraction eluted by 2.0 mol/L NaCl solutions was collected during the FPA98 ion-exchange resin elution process.

### 3.3. Preparation of LFG

LFG-Na was subjected to glycosidic bond-selective β-eliminative cleavage of the starting material NFG through its activated benzyl ester derivative according to previous research with minor modifications [12,25]. The process of LFG-Na mainly included the following steps: quaternization, carboxyl esterification, elimination and depolymerization, saponification, terminal group reduction, ultrafiltration and freeze-drying. Briefly describing the preparation process of one batch of LFG-Na (batch L3), approximately 200 g of NFG was dissolved in 2.96 L of distilled water and transalification was completed with benzethonium salts. NFG benzethonium salts were obtained by precipitation and centrifugation and dried under vacuum conditions. The NFG benzethonium salts were dissolved in 2970 mL dimethyl formamide and esterified by 63.6 mL benzyl chloride under continual stirring at 35 °C for 24 h. Then the solution was cooled to 25 °C and depolymerized by adding 1010 mL freshly prepared 0.08 mol/L EtONa in ethanol and incubating for 0.5 h. The transalification of benzethonium salt to sodium salts was completed by adding about 4.1 L saturated sodium chloride solution and 41.2 L of ethanol successively to the reaction solution. The saponification procedure in alkaline solution was necessary to hydrolyze the residual benzyl esters, and the reducing ends were reduced to its alcoholic hydroxyl by NaBH_4_ (31.9 g). After adjusting pH and precipitation by ethanol, the crude products of low molecular weight were obtained. A tangential flow ultrafiltration method with the MWCO of 3 or 10 kDa (Millipore) was selected for further purification. The fractions with $M_{w}$ higher than 3 kDa and lower than 10 kDa were collected and freeze-dried; finally, about 59.0 g LFG-Na was obtained.

### 3.4. HPSEC-MALLS-RID Analysis

#### 3.4.1. *dn/dc* Measurement of NFG and LFG-Na

The *dn/dc* value of NFG and LFG-Na were measured by using a refractive index detector (RID, Optilab rEX refractometer, DAWN EOS, Wyatt Technology Co., Santa Barbara, CA, USA). NFG or LFG-Na samples were dissolved in 0.1 mol/L sodium nitrate (containing 0.02% sodium azide), which the concentrations were about 1.8 mg/mL. From the solution, a series of solutions with six different concentrations (about 0.3, 0.6, 0.9, 1.2, 1.5, and 1.8 mg/mL) were prepared by successive dilutions. For the *dn/dc* measurements, the mobile phase was the same solvent used for the preparation of the solutions. The series of different concentrations were injected for each sample solution using a manual injector. The RID signals were analyzed and the *dn/dc* values were obtained with ASTRA software (version 7.1.3, Wyatt Technology Co., Santa Barbara, CA, USA).

##### 3.4.2. The Method of HPSEC-MALLS-RID

The $M_{w}$ and $M_{w}$/$M_{n}$ of LFG-Na and the starting material NFG were measured using HPSEC-MALLS-RID. In brief, HPSEC-MALLS-RID measurements were carried out on a multi angle laser light scattering detector (MALLS, DAWN HELEOS-II, Wyatt Technology Co., Santa Barbara, CA, USA) with an Agilent 1200 series LC/DAD system (Agilent Technologies, Palo Alto, CA, USA) equipped with a Shodex Ohpak SB-804 HQ column (300 mm × 8.0 mm) at 35 °C. The MALLS instrument was equipped with a He-Ne laser (wavelength *λ* = 661.6 nm). A refractive index detector (RID, Optilab rEX refractometer, DAWN EOS, Wyatt Technology Co., Santa Barbara, CA, USA) was simultaneously connected. The MALLS was calibrated according to the manufacturer’s recommended procedures by using HPLC grade toluene. BSA standard ($M_{w}$ of 66430 Da, *dn/dc* value of 0.185 mL/g) was employed for normalization of the 18-angle light scattering detectors relative to the right angle detector. The signals of light scattering from MALLS, refractive index from RID and UV absorbance from DAD detector were also aligned by the BSA standard. The mobile phase was 0.1 mol/L sodium nitrate (containing 0.02% sodium azide) at a flow rate of 0.5 mL/min. Each LFG-Na sample was dissolved in the same solution as the mobile phase at a final concentration of about 10 mg/mL, while NFG was about 5 mg/mL. All standard solutions and sample solutions were filtered through a 0.22 μm membrane before analysis. The Light Scattering Model was Zimm. An injection volume of 100 μL was used. The accuracy and system suitability of the method were determined via repeating the analysis of BSA for five replicates and the mean value of $M_{w}$ was compared to its labeled value (true value). The repeatability was confirmed with preparation and analysis of six parallel solutions of NFG (5 mg/mL) and LFG-Na (10 mg/mL), respectively. The intermediate precision was measured by the RSD based on a total of 12 determinations of LFG-Na on different days. The Astra software (version 7.1.3, Wyatt Technology Co., Santa Barbara, CA, USA) was utilized for data acquisition and analysis.

### 3.5. Determination of Molecular Weight and Homogeneity of LFG-Na by HPGPC

The molecular weight, including $M_{w}$, number-average molecular weight ($M_{n}$) and molecular weight distribution of LFG-Na were also determined by HPGPC using an Agilent 1200 series (Agilent Technologies, Palo Alto, CA, USA) apparatus with differential refraction detector equipped with a TSK-Gel G2000SWxl column (300 mm × 7.8 mm). According to the size-exclusion chromatography method, the chromatographic conditions and procedures are presented as follows. The measurements were carried out at 35 °C and the mobile phase was 0.1 mol/L NaCl aqueous solution at a flow rate of 0.5 mL/min. An amount of oligosaccharide standards from LFG-Na (HS5, HS8, HS11, HS14 and HS17) with mobile phase was dissolved to make the concentration 2 mg/mL, while the concentration of the test sample solution of LFG-Na was 10 mg/mL. All standard solutions and test sample solutions were filtered through a 0.22 μm membrane before analysis. An injection volume of 20 μL was used. Oligosaccharide standard retention time—the peak molecular weights curve was fitted by a third-order polynomial using Agilent GPC/SEC software, and the molecular weight and molecular weight distribution of LFG-Na were calculated using the same GPC/SEC software (version A.02.01, Agilent Technologies, Palo Alto, CA, USA).

### 3.6. Chemical Characteristics and NMR Analysis of LFG-Na

Monosaccharide composition of LFG-Na was analyzed by reverse-phase HPLC according to PMP derivatization procedures [26]. The LFG-Na sample (20 mg) was dissolved in 4 mL of 2 mol/L trifluoroacetic acid (TFA), then sealed and incubated at 110 °C for 4 h in a heating block. The reaction mixture was evaporated to dryness. The residue was then dissolved in 5 mL methanol and again evaporated to remove residual TFA for five times. The five batches of LFG-Na and the standard samples were prepared as described above. The above samples were reconstituted in 2 mL water. Then 50 μL of the sample solution, 50 μL of 0.6 mol/L sodium hydroxide and 100 μL of 0.5 mol/L PMP in methanol were mixed and incubated at 70 °C for 30 min. After adjusting the pH to 7, water was added to make the volume of solution to 1.0 mL, and then 5.0 mL of chloroform was added to extract PMP three times. The aqueous layer was collected for HPLC analysis. Analysis of the PMP-labeled polysaccharides was carried out using an Agilent technologies 1100 series LC/DAD system (Agilent Technologies, USA) and an Agilent Eclipse XDB C18 (150 mm × 4.6 mm). The flow rate was 1 mL/min, and UV absorbance of the effluent was monitored at 250 nm. Mobile phases A and B (*v/v*, 80:20) consisted of 0.1 mol/L ammonium acetate (pH 5.5) and acetonitrile, respectively.

NMR spectroscopy was performed at 298 K with a BRUKER-AVANCEIII-HD 600 MHz spectrometer equipped with a ^13^C/^1^H dual probe in FT mode according to a previously described method [27]. All samples of LFG-Na were dissolved in D_2_O and lyophilized three times to replace exchangeable protons. The lyophilized samples were then dissolved in D_2_O at a concentration of 20–30 g/L. The 1D (^1^H/^13^C) and 2D (^1^H-^1^H COSY, TOCSY, NOESY, ^1^H-^13^C HSQC, and HMBC) NMR spectra were recorded with HOD suppression by presaturation. All chemical shifts were relative to internal 3-(trimethylsilyl)-propionic-2,2,3,3-d4 acid sodium salt (TSP, *δ*_H_ and *δ*_C_ = 0.00). The NMR spectra were processed using a trial MestReNova software (v9.0.1-13254, MESTRELAB RESEARCH, S.L, Santiago de Compostela, Spain). 

### 3.7. Activity Assays of LFG-Na In Vitro

#### 3.7.1. APTT Assay

The APTT of LFG-Na was determined with a coagulometer (PRECIL C2000-4, China) using APTT kits and standard human plasma as previously described with minor modifications [23]. The LFG-Na samples were dissolved in 20 mmol/L Tris-HCl (pH 7.4) at various concentrations. LFG-Na solutions of 5 μL each were mixed with 45 μL of normal human plasma and incubated for 2 min at 37 °C. Then 50 μL of APTT reagent was added to the mixture, which was incubated for another 3 min at 37 °C. CaCl_2_ (50 μL) was then added, and the clotting time was recorded. Linear regression was performed with LFG-Na concentration as abscissa and the APTT value of LFG-Na solution at each concentration as ordinate. The concentration of LFG-Na required to double APTT was calculated according to the linear regression equation.

#### 3.7.2. Potency for Anti-iXase

The potency for anti-iXase, referred to as the PLA method, was established based on quantitative responses in the Chinese Pharmacopoeia 2020 Edition Volume IV General Principle 1431 and the methods using the reagents in the BIOPHEN FVIII:C Kit as previously described [23,27,28]. In order to compare the potency for anti-iXase among different batches of LFG-Na, sample L5 of LFG-Na was selected as the reference substance (LFG-Na RS), and the potency for its anti-iXase was set as 100 U/mg. Another four batches of LFG-Na were tested as follows. LFG-Na RS was dissolved in water and the solutions diluted 10 times with buffer solution (R4) in factor VIII kit to obtain the reference solutions with four concentrations of 0.01–0.1 U/mL (S1–S4). The test sample (T1–T4) was taken to prepare test solution in the same way. An amount of 30 μL of solutions prepared as above was added into the 96-well plate with a sequence of B1 (blank control 1), S1, S2, S3, S4, T1, T2, T3, T4, T1, T2, T3, T4, S1, S2, S3, S4, and B2. The solutions were incubated with 1 IU/mL factor VIII (50 μL), and activation reagent (R2, 50 μL) (containing human thrombin, calcium, and synthetic phospholipids) at 37 °C. The reaction was initiated by the addition of factor X (R1, 50 μL). An amount of 50 μL of factor Xa chromogenic substrate SXa-11 (R3) was added and incubated. Before detecting the absorbance (A) at 405 nm (recorded at 37 °C using a Microplate Reader (Multiskan GO-1510, Thermo Fisher Scientific, Vantaa, Finland)), 50 μL of 30% acetic acid solution was added. A was converted according to the exponential equation A′ = e^A^, linear regression was performed with A′ as the ordinate and the concentration of reference solution (or test solution) as the abscissa, respectively, and the results were input into the statistical program of Biological Assay in Chinese Pharmacopoeia BS2000 for calculating the potency and average confidence limit (FL%) of the quantitative reaction by the PLA method (4 × 4). The FL% should not be greater than 20%.

## 4. Conclusions

LFG-Na, a low molecular weight fucosylated glycosaminoglycan sodium, is a novel anticoagulant candidate that selectively inhibits iXase. In this study, the HPSEC-MALLS-RID method was successfully developed to determine the molecular weight and molecular weight distribution of LFG-Na and compared to the conventional method of HPGPC. HPSEC-MALLS-RID can also be used to determine the molecular weight of the starting material NFG. Based on multiple batches of measurement data, the acceptance criteria of the molecular weight and molecular weight distribution of NFG and LFG-Na were established. Furthermore, APTT assay was selected and the potency for anti-iXase, referring to the parallel line assay (PLA) method, was established to clarify biological potency of LFG-Na. This work illustrated that HPSEC-MALLS-RID and bioactivity assays were critical quality control methods for multi-component glycosaminoglycan LFG-Na. The methods also provide a feasible strategy to control the quality of other polysaccharide medicines.

## Figures and Tables

**Figure 1 molecules-27-04522-f001:**
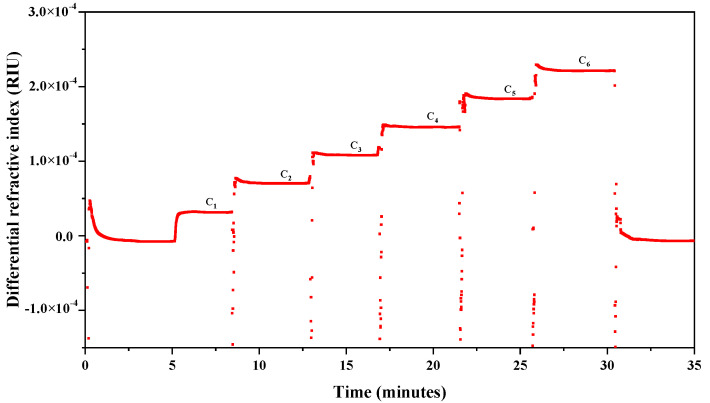
Chromatogram obtained from the determination of *dn/dc* value (LFG-Na, batch L3, concentrations of LFG-Na solutions-differential refractive index were fitted, the Fit R^2^ = 1.0000 from ASTRA software (version 7.1.3, Wyatt Technology Co., Santa Barbara, CA, USA)).

**Figure 2 molecules-27-04522-f002:**
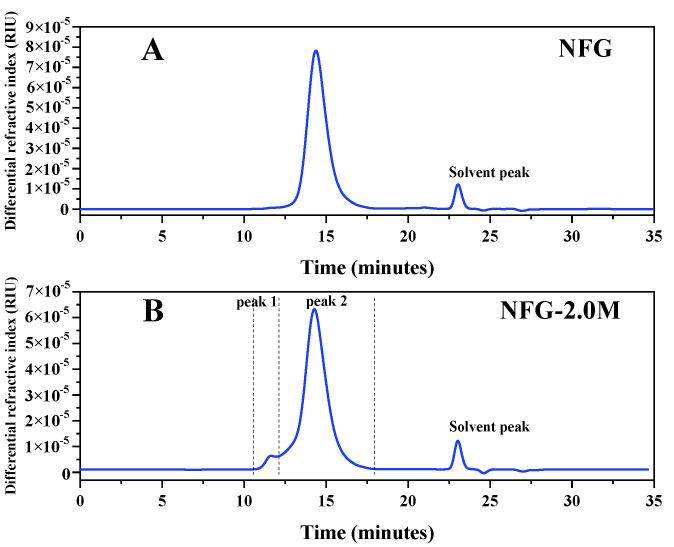
Representative HPSEC chromatogram of NFG (**A**) and NFG-2.0M (**B**) obtained before the NFG manufacturing process optimization from the determination of HPSEC-MALLS-RID.

**Figure 3 molecules-27-04522-f003:**
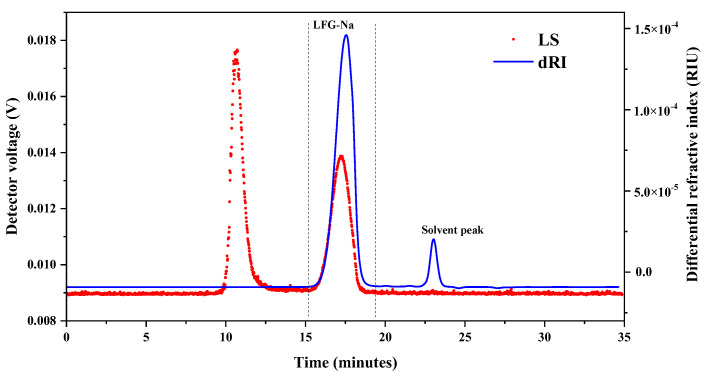
Representative HPSEC-MALLS-RID chromatogram of LFG-Na.

**Figure 4 molecules-27-04522-f004:**
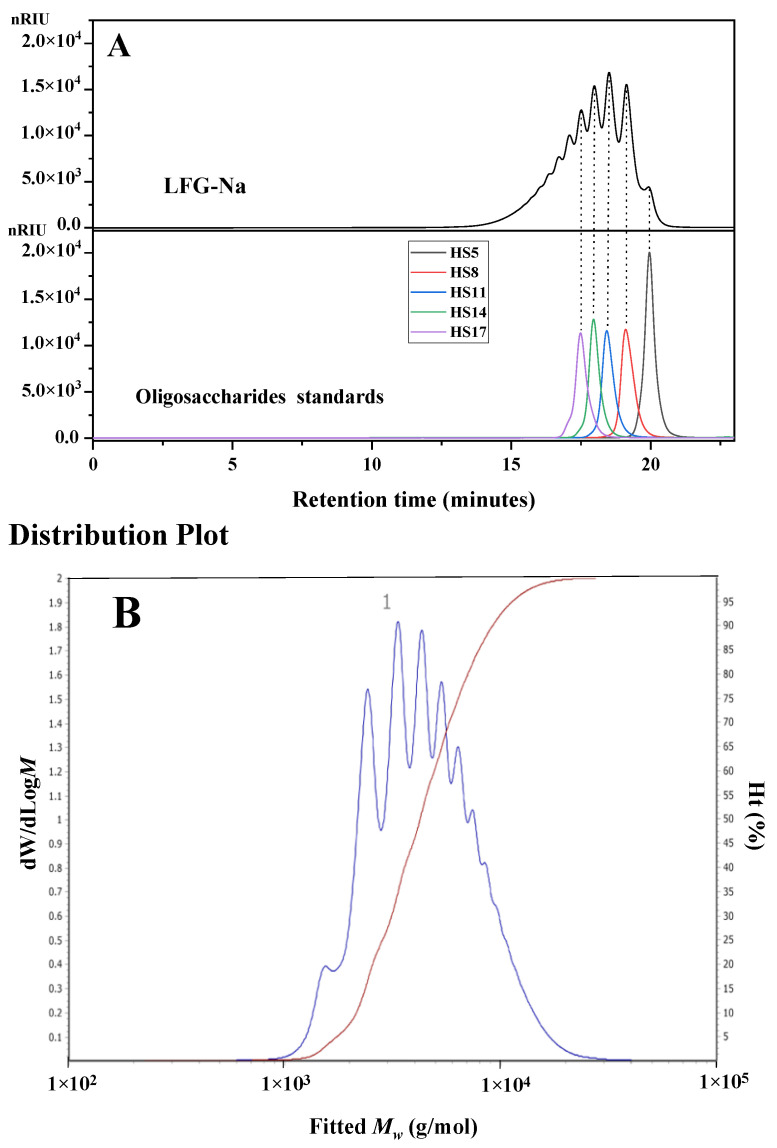
HPGPC profiles of LFG-Na and oligosaccharide standards HS5, HS8, HS11, HS14, HS17, which were determined by the differential refractive index detector (**A**). The distribution plot of LFG-Na was analyzed by using Agilent GPC/SEC software (version A.02.01, Agilent Technologies, Palo Alto, CA, USA) (**B**).

**Figure 5 molecules-27-04522-f005:**
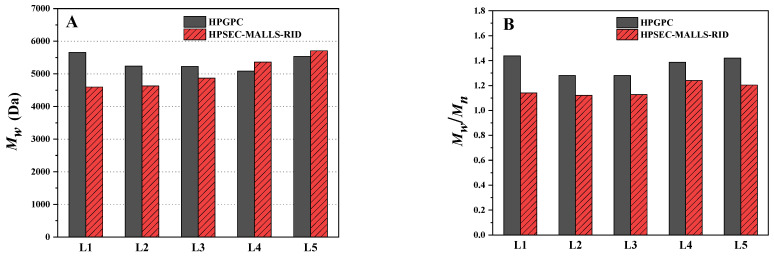
The results of molecular weight (**A**) and molecular weight distribution (**B**) for LFG-Na by HPGPC and HPSEC-MALLS-RID.

**Figure 6 molecules-27-04522-f006:**
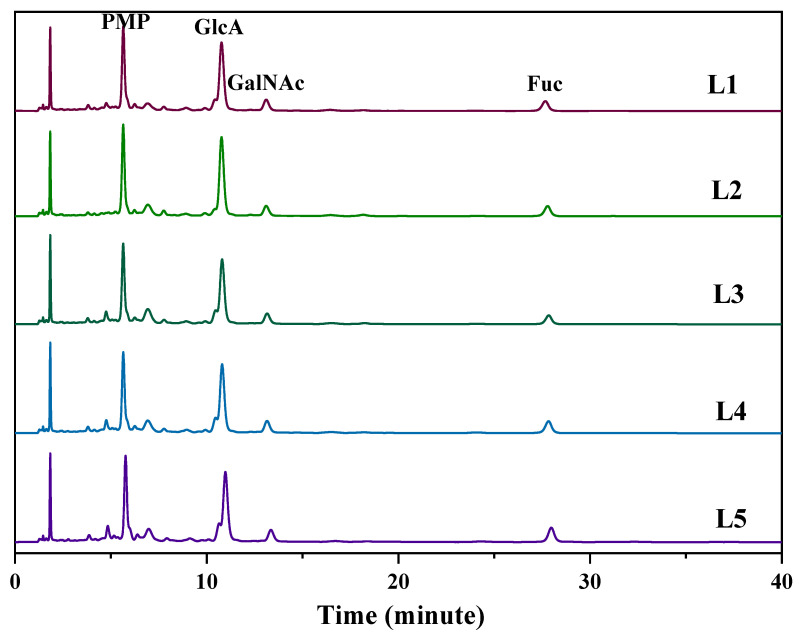
HPLC profiles of PMP derivatives from LFG-Na (L1–L5).

**Figure 7 molecules-27-04522-f007:**
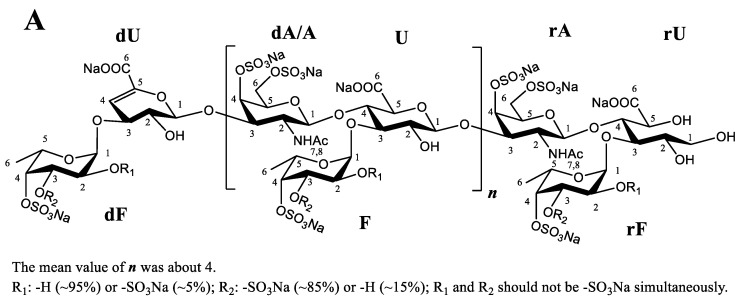
The structure of LFG-Na (**A**), 1D ^1^H NMR spectra (**B**) and ^13^C NMR spectra (**C**) from five batches of LFG-Na (L1–L5). The letters of (**A**) between parentheses are used as labels for assignments in the ^1^H-NMR (**B**) and ^13^C-NMR spectrum (**C**). They are the following: dU for the nonreducing terminal unsaturated uronic acid residue; U for internal glucuronic acid; rU for reducing terminal alcohol; dA for GalNAc_4S6S_ linked to dU; A for internal GalNAc_4S6S_; rA for GalNAc_4S6S_ linked to rU; dF for Fuc_3S4S_ residue linked to dU; F for Fuc_3S4S_ residue linked to U; rF for Fuc_3S4S_ residue linked to rU. 1D ^1^H (**B**) and ^13^C (**C**) NMR spectra are from five batches of LFG-Na (L1–L5).

**Table 1 molecules-27-04522-t001:** Measurement results of *dn/dc* values of LFG-Na and NFG using Optilab rEX refractometer.

Sample	Batch	*dn/dc* (mL/g)	Mean Value (mL/g)
LFG-Na			
	L1	0.1277	0.1248 ± 0.0046
	L2	0.1269	
	L3	0.1237	
	L4	0.1283	
	L5	0.1173	
NFG			
	N1	0.1168	0.1163 ± 0.0028
	N2	0.1154	
	N3	0.1187	
	N4	0.1113	
	N5	0.1162	
	N6	0.1192	
NFG-2.0M			
	NM1	0.1162	0.1166 ± 0.0008
	NM2	0.1172	
	NM3	0.1157	
	NM4	0.1173	

**Table 2 molecules-27-04522-t002:** The $M_{w}$ and the $M_{w}$/$M_{n}$ of NFG and NFG-2.0M by HPSEC-MALLS-RID (only the NFG peak was involved in the calculation).

Sample	Batch	$$M_{w}\;(\mathrm{Da})$$	$M_{w}$ $/M_{n}$
NFG			
	N1	60,000	1.184
	N2	70,940	1.239
	N3	61,350	1.180
	N4	58,270	1.175
	N5	74,280	1.241
	N6	64,260	1.105
NFG-2.0M			
	NM1	84,840	1.407
	NM2	86,950	1.360
	NM3	77,680	1.297
	NM4	84,320	1.360

**Table 3 molecules-27-04522-t003:** Test results of molecular weight and molecular weight distribution for LFG-Na by HPGPC and HPSEC-MALLS-RID.

Batch	HPGPC					HPSEC-MALLS-RID				
	Molecular Parameters		Molecular Weight Distribution			Molecular Parameters		Molecular Weight Distribution		
	$M_{w}\;(\mathrm{Da})$	$M_{w}$ $/M_{n}$	M1500	M1500~8000	M8000	$M_{w}\;(\mathrm{Da},\;\pm \mathrm{Error})$	$M_{w}$ $/M_{n}\;(\pm \mathrm{Error})$	M1500	M1500~8000	M8000
L1	5657	1.438	1.25%	85.34%	13.41%	4596 (±0.546%)	1.141 (±0.897%)	0	94.192%	5.808%
L2	5241	1.281	0.57%	86.80%	12.63%	4629 (±0.519%)	1.121 (±0.851%)	0	94.539%	5.461%
L3	5227	1.280	0.53%	86.60%	12.86%	4870 (±0.583%)	1.128 (±0.921%)	0	93.024%	6.976%
L4	5082	1.387	2.52%	82.91%	14.57%	5361 (±0.479%)	1.240 (±0.699%)	0	87.329%	12.671%
L5	5531	1.420	1.91%	81.48%	16.61%	5708 (±0.541%)	1.204 (±0.854%)	0	85.055%	14.945%

M1500: the percentage of the fraction of LFG-Na with molecular weights lower than 1500 Da; M1500~8000: the percentage of the fraction of LFG-Na with molecular weights range from 1500 to 8000 Da; M8000: the percentage of the fraction of LFG-Na with molecular weights greater than 8000 Da.

**Table 4 molecules-27-04522-t004:** NMR data and assignments of LFG-Na (Batch L1).

Sugar Ring	H	*δ* ^a^	^1^H-^1^H Couplings ^b^	Correlated Signals			C	*δ*	Correlated Signals	
				COSY	TOCSY	ROESY			HSQC	HMBC
rU	H-1	3.800	*J*_(1,1′)_ = 11.33	H1′, H2	H1′, H2, H3/4	H2, H3	C-1	65.40	H-1	H-1
	H-1′	3.761	*J*_(1,2)_ = 6.12	H1, H2	H1, H2, H3/4	H3			H-1′	H-1′
	H-2	4.146	*J*_(1′,2)_ = 3.96	H1/1′, H3	H1/1′, H3/4	H1/1′, H3	C-2	72.67	H-2	--
	H-3	4.072	--	H2	H1/1′, H2	H2; rF1	C-3	82.54	H-3	--
	H-4	4.052	--	H5	H1/1′, H2	--	C-4	82.54	H-4	--
	H-5	4.329	--	H4	H4	--	C-5	74.78	H-5	--
	--	--	--	--	--	--	C-6	180.22	--	H5
rF	H-1	5.116	*J*_(1,2)_ = 4.02	H2	H2, H3, H4	H2; rU3	C-1	104.33	H-1	H1, H5; rU3
	H-2	3.885	--	H1, H3	H1, H3, H4	H1, H3, H4	C-2	69.36	H-2	H2, H3, H4
	H-3	4.610	*J*_(3,4)_ =2.94	H2, H4	H1, H2, H4	H2, H3, H4, H5	C-3	78.10	H-3	H1, H2, H4
	H-4	4.909	--	H3	H1, H2, H3	H3, H5, H6	C-4	81.83	H-4	H3, H5
	H-5	4.428	--	H6	H6	H3, H4, H6	C-5	69.71	H-5	H1, H4
	H-6	1.359	--	H5	H5	H4, H5	C-6	18.94	H-6	H4, H5, H6
rA	H-1	4.695	*J*_(1,2)_ = 7.04	H2	H2, H3	H2, H3, H5; rU4	C-1	104.33	H-1	rU4
	H-2	3.974	--	H1, H3	H1, H3, H4	H3, H1	C-2	54.22	H-2	H3, H4
	H-3	4.173	--	H2, H4	H1, H2, H4	H1, H2, H4; U1	C-3	78.74	H-3	H1, H4; U1
	H-4	4.829	--	H3	H2, H3, H5	H3, H5, H6	C-4	79.25	H-4	H3, H5, H6′
	H-5	4.092	--	H6, H6′	H4, H6/6′	H1, H4, H6/6′	C-5	75.06	H-5	H6/6′
	H-6	4.276	--	H5, H6′	H5, H6′	H4, H5, H6′	C-6	70.52	H-6/6′	H5
	H-6′	4.199	--	H5, H6	H5, H6	H4, H5, H6	C-7	177.97	--	H2, H8
	H-8	2.090	--	--	--	--	C-8	25.26	H-8	H8
U	H-1	4.491	*J*_(1,2)_ = 8.88	H2	H2, H3, H4, H5	H2, H3, H5; rA/A3	C-1	106.71	H-1	H2, H3; A3
	H-2	3.616	*J*_(2,3)_ = 7.74	H1, H3	H1, H3, H4, H5	H4, H1	C-2	76.45	H-2	H2, H3
	H-3	3.690	--	H2, H4	H1, H2, H4, H5	H1, H2, H4; F1	C-3	82.00	H-3	H2, H4, H5; F1
	H-4	4.021	--	H3, H5	H1, H2, H3, H5	H2, H3/5; A1	C-4	78.07	H-4	H3, H5
	H-5	3.706	--	H4	H1, H2, H3, H4	H1, H3	C-5	79.84	H-5	H4, H5
	--	--	--	--	--	--	C-6	177.87	--	H4, H5
A	H-1	4.485	*J*_(1,2)_ = 8.52	H2	H2, H3	H2, H3, H5; U4	C-1	102.34	H-1	H2; U4
	H-2	4.021	--	H1, H3	H1, H3, H4	H3, H1	C-2	54.24	H-2	H3, H4
	H-3	3.895	--	H2, H4	H1, H2, H4	H1, H2, H4; U1	C-3	79.11	H-3	H1, H4; U1
	H-4	4.751	--	H3	H2, H3, H5	H3, H5, H6	C-4	78.90	H-4	H3, H5, H6′
	H-5	3.932	--	H6, H6′	H4, H6/6′	H1, H4, H6/6′	C-5	74.62	H-5	H6/6′
	H-6	4.287	--	H5, H6′	H5, H6′	H4, H5, H6′	C-6	69.88	H-6/6′	H5
	H-6′	4.183	--	H5, H6	H5, H6	H4, H5, H6	C-7	177.84	--	H2, H8
	H-8	1.998	--	--	--	--	C-8	25.38	H-8	H8
F	H-1	5.356	*J*_(1,2)_ = 3.72	H2	H2, H3, H4	H2, H3, H4, H5; U3	C-1	102.06	H-1	H3; U3
	H-2	3.950	*J*_(2,3)_ = 10.08	H1, H3	H1, H3, H4	H1, H3, H4	C-2	69.24	H-2	H2, H3, H4
	H-3	4.521	--	H2, H4	H1, H2, H4	H2, H3, H4, H5	C-3	78.28	H-3	H1, H2, H4
	H-4	5.040	--	H3	H1, H2, H3	H3, H5, H6	C-4	82.20	H-4	H3, H5
	H-5	4.849	--	H6	H6	H3, H4, H6	C-5	69.28	H-5	H1, H4
	H-6	1.404	--	H5	H5	H4, H5	C-6	18.89	H-6	H4, H5, H6
dA	H-1	4.571	*J*_(1,2)_ = 8.52	H2	H2, H3	H2, H3, H5, U4	C-1	102.51	H-1	H2; U4
	H-2	4.163	--	H1, H3	H1, H3, H4	H3, H1	C-2	54.37	H-2	H3, H4
	H-3	4.166	--	H2, H4	H1, H2, H4	H1, H2, H4, dA1	C-3	78.69	H-3	H1, H4; dU1
	H-4	4.988	--	H3	H2, H3, H5	H3, H5, H6	C-4	78.91	H-4	H3, H5, H6′
	H-5	4.088	--	H6, H6′	H4, H6/6′	H1, H4, H6/6′	C-5	74.99	H-5	H6/6′
	H-6	4.371	--	H5, H6′	H5, H6′	H4, H5, H6′	C-6	70.16	H-6/6′	H5
	H-6′	4.263	--	H5, H6	H5, H6	H4, H5, H6	C-7	178.04	--	H2, H8
	H-8	2.065	--	--	--	--	C-8	25.38	H-8	H8
∆U	H-1	4.922	*J*_(1,2)_ = 7.20	H2	H2, H3, H4	H1, H3, H4, dA3	C-1	105.91	H-1	H1, H2; dA3
	H-2	3.911	*J*_(2,3)_ = 8.22	H1, H3	H1, H3, H4	H3	C-2	73.10	H-2	H2, H3, H4
	H-3	4.491	--	H2, H4	H1, H2, H4	H1, H2, H4, dF1	C-3	79.90	H-3	H2; dF1
	H-4	5.760	--	H3	H1, H2, H3	H1, H3	C-4	109.54	H-4	H3, H4
	--	--	--	--	--	--	C-5	149.79	--	H4, H3
	--	--	--	--	--	--	C-6	171.76	--	H4
dF	H-1	5.282	*J*_(1,2)_ = 3.96	H2	H2, H3, H4	H2, dU3	C-1	101.15	H-1	H5; dU3
	H-2	3.951	*J*_(2,3)_ = 10.88	H1, H3	H1, H3, H4	H1, H3, H4	C-2	69.24	H-2	H2, H3, H4
	H-3	4.610	*J*_(3,4)_ = 3.06	H2, H4	H1, H2, H4	H2, H3, H4, H5	C-3	78.10	H-3	H1, H2, H4
	H-4	4.909	--	H3	H1, H2, H3	H3, H5, H6	C-4	81.83	H-4	H3, H5
	H-5	4.356	--	H6	H6	H3, H4, H6	C-5	70.03	H-5	H1, H4
	H-6	1.301	--	H5	H5	H4, H5	C-6	18.71	H-6	H4, H5, H6

a *δ*: Chemical shift, ppm, internal standard: deuteration TSP. b *J*: Coupling constant, Hz.

**Table 5 molecules-27-04522-t005:** Bioactivities and molecular weights of LFG-Na.

Batch	$M_{w}{\;}^{a}\;(\mathrm{Da})$	APTT ^b^ (μg/mL)	Anti-iXase ^c^ (U/mg)
L1	4596	8.89 ± 0.08	88.7
L2	4629	8.84 ± 0.03	85.9
L3	4870	8.64 ± 0.06	89.8
L4	5361	7.92 ± 0.14	94.1
L5	5708	7.36 ± 0.11	100

^a^ $M_{w}$ data are from the result of HPSEC-MALLS-RID. ^b^ Concentrations required to double the clotting time (*n* = 2). ^c^ Sample L5 of LFG-Na was selected as the reference substance, and the potency for anti-iXase was set as 100 U/mg.

## Data Availability

Not applicable.

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
