# Peer review of "Critical Quality Control Methods for a Novel Anticoagulant Candidate LFG-Na by HPSEC-MALLS-RID and Bioactivity Assays"

_molecules, 2022, doi:10.3390/molecules27144522_

Round 1

Reviewer 1 Report

Zheng and co-workers present their studies on the development of quality control methods for an oligosaccharide-based anticoagulant candidate LTF-Na. Through chemical degradation of fucosylated glycosaminoglycan from see cucumber Holothuria fuscopunctata Jaeger, a mixture of low molecular weight fucosylated glycosaminoglycan sodium (LFG-Na) were obtained and identified as a novel anticoagulant candidate to enter clinical trials. The determination of the molecular weight and molecular weight distribution of such a mixture oligosaccharides is challenging but critical for quality control process. In this report, they applied high performance size exclusion chromatography coupled with multi angle laser light scattering and refractive index detector (HPSEC-MALLS-RID) to evaluate the molecular weight and distribution of different batches of LFG-Na, which were further supported by NMR analysis and activated partial thromboplastin time assay.

The paper is very well written and describes a very useful workflow to control the quality of oligosaccharide drugs. Experiment are well conducted and relatively convincing. It warrants publication.

Author Response

We appreciate your recognition and affirmation for our study.

Thank you and best regards.

Reviewer 2 Report

Major revision comments

In general, the study is precise in the working methodology, analytical design and the presentation of results. The comments and the analytical results obtained are accurate and focused on the potential usage of low molecular weight fucosylated glycosaminoglycan sodium (LFG-Na) from the sea cucumber Holothuria fuscopunctata as a novel anticoagulant candidate.

1.      Abstract

a.      Background is missing in the abstract.

b.      The aim of the study should be clearly specified in the abstract. It is not clear what the novelty of this study is.

2.      Materials and methods:

Line 422 - 444: It was not clear from the paragraph regarding HPSEC-MALLS-RID, is this method validated according to validation criteria? Only the repeatability was mentioned to be established. What are the accuracy, the intermediate precision, and the recovery of the method?

3.      Results and Discussion:

The strengths and the limitations of the study should be specified.

Author Response

Thank you very much for the valuable comments and suggestions. We have revised our manuscript carefully and the modifications are marked up using the 
“Track Changes” function. In what follows, we reply to your comments point by point.

Point 1: Abstract

  1. Background is missing in the abstract.
  2. The aim of the study should be clearly specified in the abstract. It is not clear what the novelty of this study is.

Response 1: Thank you for the comment. To reflect the relevant information of Background and clearly specify the aim and novelty of the study, we have added the description of “The molecular weight, molecular weight distribution and bioactivities are the critical quality attributes of LFG-Na. The determination of these quality attributes of such an oligosaccharides mixture drug is challenging but critical for quality control process to ensure the safety and efficacy in clinical use” to the Abstract (Line 14-17).

Point 2: Materials and methods 

Line 422 - 444: It was not clear from the paragraph regarding HPSEC-MALLS-RID, is this method validated according to validation criteria? Only the repeatability was mentioned to be established. What are the accuracy, the intermediate precision, and the recovery of the method?

Response 2:

(1) Yes, this method is validated according to validation criteria.

(2) Thank you for the comment, the accuracy and the intermediate precision of HPSEC-MALLS-RID method have been added to section 3.4.2(Line 455-460). Besides, the results of accuracy and the intermediate precision also have been added to section 2.2 (Line 133-138) and 2.3 (Line 186-192).

(3) The purpose of HPSEC-MALLS-RID method is to determine the molecular weight and molecular weight distribution, not the analyte concentration, therefore, not all of the performance parameters for HPLC method validation are applicable. In the authors’ opinion, the recovery generally does not need to be determined in this HPSEC-MALLS-RID method.

Point 3: Results and Discussion:

The strengths and the limitations of the study should be specified. 

Response 3:

Thank you for the comment. The strengths of the study have been described in our article relatively well. Therefore, we focus on adding the description of the limitations. The description of “The limitation of HPSEC-MALLS-RID method was that the dn/dc value of the sample should be determined or obtained before determining the molecular weight and molecular weight distribution. In addition, the accuracy and system suitability of the method should be assessed by standard, such as BSA or Dextran, which was used at the beginning of each run sequence” was added to section 2.5(Line 268-273).

Thank you for your constructive suggestions on this paper. Please let us know if you have any requirement for further revision. We would like to try our best to address the questions raised as soon as possible.

Thank you and best regards.